# Dialdehyde Cellulose Solution as Reducing Agent: Preparation of Uniform Silver Nanoparticles and In Situ Synthesis of Antibacterial Composite Films with High Barrier Properties

**DOI:** 10.3390/molecules28072956

**Published:** 2023-03-26

**Authors:** Jinsong Zeng, Xinyi Xiong, Fugang Hu, Jinpeng Li, Pengfei Li

**Affiliations:** 1Plant Fibril Material Science Research Center, State Key Laboratory of Pulp and Paper Engineering, School of Light Industry and Engineering, South China University of Technology, Guangzhou 510640, China; 2Guangdong Provincial Key Laboratory of Plant Resources Biorefinery, Guangzhou 510006, China; 3School of Environment and Energy, South China University of Technology, Guangzhou 510640, China

**Keywords:** dialdehyde cellulose solution, silver nanoparticles, in situ synthesis, antibacterial film

## Abstract

The demand for antimicrobial materials is gradually increasing due to the threat of infections and diseases caused by microorganisms. Silver nanoparticles (AgNPs) are widely used because of their broad-spectrum antimicrobial properties, but their synthesis methods are often environmentally harmful and AgNPs difficult to isolate, which limits their application in several fields. In this study, an aqueous solution of dialdehyde cellulose (DAC) was prepared and used as a reducing agent to synthesize AgNPs in an efficient and environmentally friendly process. The synthesized AgNPs can be easily separated from the reducing agent to expand their applications. In addition, the AgNPs were immobilized in situ on dialdehyde cellulose to form antibacterial composite films. The results showed that the prepared silver nanoparticles were mainly spherical and uniformly dispersed, with an average size of about 25 nm under optimal conditions. Moreover, the dialdehyde cellulose–nanosilver (DAC@Ag) composite films had excellent mechanical properties, positive transparency, ultraviolet-blocking properties, and effective antibacterial activity against *E. coli* and *S. aureus*. Notably, the composite films exhibited excellent oxygen and water vapor barrier properties, with WVT and ORT of 136.41 g/m^2^·24 h (30 °C, 75% RH) and <0.02 cm^3^/m^2^·24 h·0.1 MPa (30 °C, 75% RH), respectively, better than commercial PE films. Hence, this study not only provides an environmentally friendly method for the preparation of silver nanoparticles, but also offers a simple and novel strategy for the in situ synthesis of silver-loaded antibacterial composite films.

## 1. Introduction

In recent decades, with the improvement in health awareness, people have gradually recognized the hazards of illness transmission provoked by microorganisms. As a result, the usage of antimicrobial materials has risen in a variety of applications, including medical hemostatic antibacterial dressings, antimicrobial packaging materials, etc. [1,2,3]. In addition to the commonly used antibiotics, some natural or synthetic organic substances have been used as antimicrobial agents due to their excellent antimicrobial properties [4,5], such as cinnamon oil, chitosan, and quaternary ammonium salts [6,7,8]. However, the selection of appropriate antibacterial substances is still a huge challenge, due to poor heat resistance and easy-to-produce drug resistance [9]. As an antibacterial material with excellent performance, AgNPs not only exhibit broad antimicrobial activity against a wide range of pathogens such as bacteria, viruses, and fungi, but also have low cytotoxicity [10], and no microbial resistance, which has expanded their applications in medicine [11], antimicrobial packaging [12], material coatings [13], and water treatment [14]. Recently, many technologies have been developed in improving the ability to produce AgNPs as a result of great efforts devoted to thermal decomposition, UV irradiation, chemical reduction, laser etching, γ irradiation, etc. [15,16,17,18]. Among them, one of the most prevalent approaches has been the use of efficient reducing agents for chemical reduction, including borohydride, hydrazine and other substances, which have caused great damage to the environment [19]. In addition, it is hard to separate AgNPs from these added chemicals completely, limiting the applicability of generated AgNPs [20]. Therefore, it is still highly desirable to explore facile and green methods for synthesizing AgNPs with excellent antibacterial properties.

As the most abundant natural resource on Earth, cellulose has become a hot biological material with great development potential due to its biocompatibility, regeneration, biodegradability, low cytotoxicity, and environmental friendliness [21]. To date, a great deal of research has been devoted to the development of polymers containing cellulose, including nanocellulose and cellulose derivatives, and cellulose has been used in reinforcing materials, oil–water separation, food packaging, medical materials, etc. [22,23,24,25]. Antimicrobial nanoparticles combined with oxidized cellulose can reduce unwanted effects such as allergies and inflammation; moreover, the shapes and sizes of antimicrobial nanoparticles can be altered [26,27]. It has been broadly reported that a variety of oxidized celluloses can be used to prepare biocompatible products. In particular, dialdehyde cellulose, as an intriguing material, can form primary amines, secondary amines, or Schiff bases, offering a large number of possibilities for further chemical modifications, including attachment to antimicrobial units [28]. In addition, dialdehyde cellulose with a high oxidation degree (>90%) can form a homogeneous solution in hot water (>80 °C) and can be used as an environmentally friendly reducing agent [29]. The active sites at the C2 and C3 positions of the glucose unit on the surface of cellulose modified by oxidant can effectively restrict the reaction process due to the steric hindrance effect. Xiao et al. obtained ultrafine gold nanoparticles using dialdehyde cellulose as an efficient reducing agent [30]. However, to the best of our knowledge, the preparation of AgNPs using a homogeneous solution of dialdehyde cellulose or the in situ synthesis of DAC-AgNP regenerated composite membranes on it has not been reported.

In this study, highly oxidized dialdehyde cellulose was dissolved in hot water to form a homogeneous solution, which was then used as a reducing agent to prepare AgNPs under oil bath heating and stirring. The reaction process was convenient and efficient. Moreover, although this reaction originated from a simple Tollens reaction, we could not only easily isolate the synthesized AgNPs individually, but also successfully immobilized them in situ on dialdehyde cellulose to form a regenerated antibacterial composite film. The effects of different reaction conditions on the synthesis of AgNPs were studied, and the structural changes of the materials during the reaction were analyzed. In addition, we also prepared in situ-synthesized DAC@Ag composite films and analyzed their micromorphology, elemental analysis, mechanical properties, oxygen and water vapor barrier, water contact angle, light transmittance, and antibacterial properties. This study provides insight into the design of packaging materials with excellent antibacterial capacity by exploiting a new green controllable reduction method.

## 2. Results and Discussion

Previous studies have shown that the antibacterial effect of AgNPs is related to shape and particle size, with triangular AgNPs having the highest antibacterial activity, followed by spherical AgNPs [31], and in addition, the smaller the particle size the, better the antibacterial effect [32,33]. So here, we explored the shape and particle size of AgNPs prepared by using DAC solution as reducing agent and discussed the effect of reaction conditions to determine the optimal conditions to control their particle size range.

### 2.1. Effect of Reaction Conditions on the Formation of Silver Nanoparticles

#### 2.1.1. Different Mass Ratio of DAC to Silver Ammonia Solution

The surface plasmon resonance (SPR) of nanosilver colloids reflected the behavior of silver nanoparticles [34], as assessed by UV-vis spectroscopy. In Figure 1a, the surface plasmon resonance (SPR) in the UV-visible spectrum was distributed at about 420 nm, indicating the formation of AgNPs [35]. Furthermore, the symmetrical plasmonic band shapes indicated that the AgNPs were spherical and monodisperse [36]. As displayed in Figure 1a, with the increase in the mass ratio of DAC to silver ammonia solution, the UV peaks became sharper, indicating a more uniform particle size distribution [37]. It may be because of the weak reduction ability and the poor reduction effect under the reduced amount of DAC, so the generated particles were not uniform, resulting in a wide particle size distribution of these AgNPs. However, with the increase in DAC mass ratio, the reduction ability became stronger and the reduction effect was better, so the size distribution was more uniform. In addition, the absorbance of the spectra was enhanced with increasing mass ratio of DAC to silver ammonia solution, indicating more nanosilver generation. With the decrease in the mass ratio of DAC to silver ammonia solution, the peak appeared red-shifted, indicating AgNP aggregation or an increase in the average size of AgNPs [38]. Moreover, the average size histogram (Figure 1a, inset) showed that the average size of AgNPs was smallest when the mass ratio of DAC to silver ammonia solution increased to 2 or 4, which is consistent with the results of UV-vis spectroscopy analysis. Figure 1b–f display the TEM images of the AgNPs, in which the AgNPs were almost spherical in shape, and the AgNPs with the mass ratio for DAC to silver ammonia solution of 2 or 4 had the smallest particle size. Overall, the mass ratio of DAC to silver ammonia solution of ≥2 was considered to be the best ratio condition.

#### 2.1.2. Different Reaction Times

As shown in the UV-vis spectra of Figure 2a, the reaction time had little effect on the half-peak width of the curves, indicating a uniform size distribution of silver nanoparticles [19]. In addition, as the reaction time increased, the UV absorbance rose, indicating that more AgNPs were formed during the reaction. However, when the reaction time was increased to 100 min, a slight red shift of the peak could be seen, probably due to the slight aggregation of the particles caused by the long reaction time. The results for the average size histogram (Figure 2a, inset) showed that the particle sizes of AgNPs obtained at different reaction times were similar, with the average particle size around 24–26 nm, indicating that the reaction time had almost no effect on the preparation of AgNPs. Figure 2b–f show the TEM images of the AgNPs obtained at different reaction times, in which the particle sizes were almost the same due to the stable reaction for the formation of AgNPs. In addition, the TEM of Figure 2d showed a more uniform distribution. Therefore, the best reaction time of 60 min was determined based on the above analysis.

#### 2.1.3. Different Reaction Temperatures

As shown in Figure 3a, the UV-vis spectra of different reaction temperatures exhibited similar half-peak widths. Meanwhile, the peaks of the spectra had a slight red shift with the increasing reaction temperature, indicating the aggregation or increase in particle size. This may have been due to the increase in temperature exacerbating Brownian motion, making the particles more prone to collision and aggregation [39]. The average size histogram in the inset showed that as the reaction temperature increased, there was an increasing tendency for the average particle size of AgNPs, indicating the aggregation of particles. As shown in Figure 3b–f, the generated AgNPs were almost spherical in shape, and the size of the AgNPs slightly increased with the increase in the reaction temperature. Additionally, the particles were most uniformly distributed at 50 °C. Therefore, the optimal reaction temperature of 50 °C was determined via the above analysis.

#### 2.1.4. Different Silver Ion Contents

Combining the above experimental results, we also investigated the case of different Ag^+^ content (wt%). As shown in the UV-vis spectra of Figure 4a, the absorbance rose with the increase in Ag^+^ content, which was because the increase in Ag^+^ content produced more AgNPs. However, the absorbance decreased, and the half-peak width broadened when the content was above 0.1 wt%, which might be attributed to the increase in Ag^+^ content leading to more AgNP generation, increased collision probability and partial AgNP aggregation and precipitation [19]. As shown in the average size histogram (Figure 4b), the average particle sizes of AgNPs produced with Ag^+^ content of 0.05 wt% to 0.1 wt% were similar, in the range of 25–29 nm. However, when the Ag^+^ content continued to increase, the average particle size increased rapidly, which may have been due to the aggregation of large AgNPs. Figure 4c–e show the TEM images of AgNPs at the first three Ag^+^ contents, in which the particle sizes and shapes of all three were similar. Combined with the above analysis, it can be seen that the particle sizes of the AgNPs were almost independent of the Ag^+^ content in a certain range of Ag^+^ content.

Combining the different analyses of each condition above, it can be seen that when the homogeneous solution of DAC was used as the reducing agent, AgNPs with a particle size of about 25 nm could be prepared steadily under heating conditions. In the best synthesis conditions, the mass ratio of DAC to silver ammonia solution was ≥2, the reaction time was 60 min, the reaction temperature was 50 °C, and the Ag^+^ content was ≤0.1 wt%.

### 2.2. Chemical Structure and Morphological Analysis

Figure 5a shows the FT-IR spectra of the original MCC and DAC. The characteristic absorption band of MCC at 3363 cm^−1^ was attributed to the O-H stretching vibration, the vibrational absorption peak near 2895 cm^−1^ was caused by the symmetric C-H vibration, and the absorption peak at 1024 cm^−1^ was attributed to the C-O-C pyranose ring skeleton vibration. The new absorption band of DAC at about 1735 cm^−1^ was attributed to the C=O stretching vibration of the free aldehyde in the oxidized aldehyde cellulose, and the new absorption peak at 877 cm^−1^ was attributed to the hemiacetal structure [29]. As shown in Figure 5b, the XRD patterns of MCC showed diffraction peaks at 2θ approximately equal to 14.8°, 16.2°, 22.6°, and 34.6°, corresponding to the (101), (10-1), (002), and (040) crystallographic planes in the typical cellulose I crystal form, respectively [40]. The crystallinity of MCC was calculated to be 81.16% using the Jade 6.5 treatment and equation (1). In addition, the highly oxidized DAC (DO ≥ 90%) has only a broad peak at 2θ approximately equal to 18.9°, and the rest of the diffraction peaks disappeared because the crystalline structure of MCC was highly disrupted during the NaIO_4_ oxidation. The degree of oxidation of DAC in this study was as high as 90% (over 87%), so it became completely amorphous material [41]. Figure 5d shows the XRD spectra of the AgNPs produced by the reaction under optimal conditions. The diffraction peaks appearing at 2θ approximately equal to 38.2°, 44.42°, 64.54°, 77.48°, and 81.68° corresponded to the (111), (200), (220), (311) and (222) crystallographic planes of Ag, respectively, which was agreement with the standard value JCPDS (no. 89-3722) [42]. The high-resolution transmission electron microscopy in Figure 5c showed that the generated nanosilver had a regular lattice, and the measured crystalline surface spacing was 0.236 nm, which was consistent with the standard (200) crystalline surface spacing of nanosilver. The inset of Figure 5c shows the selected electron diffraction of nanosilver. The appearance of speckle diffraction rings indicated that the generated nanosilver had good crystallinity and was polycrystalline in structure. According to the Ag XRD spectra by Jade 6.5 with equation (2), the average size of AgNPs was calculated to be about 23 nm, being similar to the results of Malvern analysis for average particle size. Figure 5e depicts the full scan XPS spectra (0–1200 eV) of DAC and the generated AgNPs. In addition to C and O, the spectra of AgNPs also contained Ag, indicating the successful preparation of AgNPs. The peaks of Ag3d_5/2_ and Ag3d_3/2_ in the inset appeared at 367.3 and 373.3 eV, confirming the reduction of Ag1 to Ag0 during the hydrothermal reaction [43].

### 2.3. Performance Analysis of DAC@Ag Composite Films Synthesized In Situ

Figure 6a shows the optical photographs of DAC@Ag0-DAC@Ag5 films, in which all the films had good transparency and darkened with increasing silver content. Figure 6b,c show the SEM images of the planar and cross-sectional DAC@Ag0 films in which the surface of the samples was very smooth and dense, and the fracture edges of the cross-section had slight irregularities. This proved that there were no pores in the DAC films, which is consistent with the results of existing study [28]. As shown in Figure 6d,e, the planar SEM revealed the dense surface of the DAC@Ag1 film and the AgNPs generated in situ on the film, while the cross-sectional SEM was similar to that of the DAC@Ag0 film. Figure 6f,g show the EDS elemental mapping of the composite film planes and cross sections. The results show a homogeneous distribution of silver in the DAC@Ag composite films, indicating that the in situ generated AgNPs were uniformly distributed throughout the films.

Appendix A shows the XRD spectra of the DAC@Ag0-DAC@Ag5 composite films. There was only one peak in the DAC@Ag0 film, appearing at about 2θ = 20°, attributed to the diffraction peak of DAC. In contrast, a new peak at about 2θ = 38°, corresponding to the (111) crystal plane of Ag, was clearly seen in the DAC@Ag1-DAC@Ag5 composite films, and this diffraction peak was gradually enhanced with the increase in the addition of silver ammonia solution. In addition, the XRD pattern of DAC@Ag5 composite film was enlarged, as shown in Appendix A. The diffraction peaks of AgNPs were shown at 38.12°, 44.24°, 64.54°, 77.33° and 81.46°, which corresponded to the (111), (200), (220), (311) and (222) planes, respectively. These characteristic peaks were consistent with the standard powder diffraction peak data for AgNPs (JCPDS, No. 89-3722), demonstrating the successful synthesis and immobilization of Ag NPs on DAC via this environmentally friendly method [44].

X-ray photoelectron spectroscopy was performed on the DAC@Ag0-DAC@Ag5 composite films to analyze their elemental composition. Appendix A shows the full scan spectra (in the range of 0–1200 eV) of the DAC@Ag0-DAC@Ag5 composite films. Apparently, two peaks appeared in the XPS spectrum of DAC@Ag0, attributed to C1s and O1s, respectively, while another different peak appeared in the spectrum of DAC@Ag1-DAC@Ag5 composite films, attributed to Ag3d of nanosilver, further confirming the successful in situ generation of nanosilver in the film.

In addition, the curve fitting to the high-resolution spectrum of XPS was performed to further analyze the chemical bonding information. Appendix A show the peak-fitted high-resolution spectra of C1s and O1s in the DAC@Ag composite films (the binding energies were calibrated with C1s = 284.6 eV). The binding energies with peaks at 284.89, 286.81 and 288.78 eV corresponded to C-C/C-H bond, C-O bond and C=O bond, respectively. Notably, the fitted peaks of C=O bond in DAC@Ag1-DAC@Ag5 composite films were clearly shifted to lower binding energies compared to DAC@Ag0. As shown in Appendix A, the curve fitting of O1s in DAC@Ag0 films showed that the peaks at 530.68, 531.37 and 532.06 eV corresponded to C=O, C-O and O-H, respectively, while the O1s peak positions of the DAC@Ag1-DAC@Ag5 composite films changed, indicating that the generated AgNPs interacted with the aldehyde group of DAC. Appendix A shows the fitted spectra of Ag3d for DAC@Ag composite films with different silver contents. It can be clearly seen that the characteristic peaks of Ag3d were double peaks located at 367 eV and 373 eV, belonging to Ag3d_5/2_ and Ag3d_3/2_, respectively. Moreover, the splitting energies of all Ag3d were about 6.0 eV, indicating that the silver ions were reduced to Ag0 by DAC [45].

Thus, the binding energy shifts of C 1s, O 1s and Ag 3d in DAC@Ag films demonstrated the strong interaction between DAC and AgNPs in DAC@Ag composite films. The above results confirmed the stable deposition of AgNPs in DAC@Ag composite films, which is consistent with previous studies [26,46].

As shown in Figure 7a, the infrared characteristic peaks of the films did not vary with the addition of silver ammonia solution. In addition, the absorption peak at 1620 cm^−1^ was attributed to the symmetric stretching vibration of the carbonyl group [26], and the absorption peak at 879 cm^−1^ was attributed to the hemiacetal structure.

Mechanical properties are one of the important characteristics of composite films [47], and the stress–strain curves of the films are displayed in Figure 7b. It can be seen that the elastic modulus of the nanosilver-loaded composite films was higher than that of the DAC@Ag0 film, which was due to the loading of nanosilver increasing the stiffness of the films. As shown in Figure 7c, the tensile strength of DAC@Ag1 and DAC@Ag2 films was higher than that of the DAC@Ag0 films, due to the AgNPs enhancing the stress transfer of those films to some extent [48,49]. However, the tensile strength of DAC@Ag3-DAC@Ag5 films was lower than that of the DAC@Ag0 films, because the filling of nano-silver weakened the hydrogen bonds between fibers [50]. Moreover, the elongation at the break of the silver-loaded composite films gradually decreased compared with the DAC@Ag0 film, which was also caused by the weakening of hydrogen bonds due to nano-silver filling. In addition, Table 1 lists the elastic modulus E, tensile strength σ, and elongation at break ε of the DAC@Ag1 films, which were 2.99 GPa, 94.07 MPa and 4.17%, respectively, and better than the reported values for cellulose–nanosilver composites (E = 3.7 GPa, σ = 54.2 MPa, ε = 4.8%) [51].

Figure 7d shows the UV-vis transmittance spectra of DAC@Ag0-DAC@Ag5 films. It is clear that DAC@Ag1 film had a high transmittance (91%) between 600 nm and 800 nm, which was similar to that of DAC film and higher than that of the cellulose composite film (84.4% at 600 nm) in other study [52]. However, the transmittance gradually decreased as the nanosilver content increased, due to the scattering between DAC and AgNPs. The downward peaks of the curves around 420 nm were attributed to the surface plasmon resonance of the nanosilver, indicating the generation of nanosilver. In addition, the composite films had lower transmittance than the DAC@Ag0 films in the UV range, and the UV transmittance decreased with the increase in the nanosilver content. These indicated that the composite film had good UV-blocking properties, which is one of the important characteristics of active packaging materials [53].

The barrier is an important feature for packaging materials, being mainly manifested by water vapor transmission rate (WVT) and oxygen transmission rate (OTR). Typically, packaging film materials require lower WVT and OTR. Figure 8a and Table 1 show that the WVT values of DAC@Ag0-DAC@Ag5 films were 152.84, 140.41, 140.20, 139.91, 137.82, and 136.41 g/m^2^·24 h, respectively, which were lower than the values for commercial PE films (3800 g/m^2^·24 h) and silver-loaded composite films in existing studies (210 g/m^2^·24 h) [42]. In addition, the WVT of the nanosilver-loaded composite film was reduced compared to that of the DAC@Ag0 film, probably due to the filling effect of AgNPs and the increase in the zigzag path [54]. As shown in Table 1, both DAC@Ag0-DAC@Ag5 films had a very low oxygen transmission rate (OTR), being below the detection limit of the instrument (0.02 cm^3^/m^2^·24 h·0.1 MPa), and a similar situation has been reported [28]. The very low oxygen transmission rate makes the films highly attractive for packaging applications.

The water stability test was measured by the one-hour water absorption of the films. As shown in Figure 8b and Table 1, the water absorption of the DAC@Ag0 film was as high as 250%, while that of the silver-loaded composite films decreased to below 4% (Figure 8b, Table 1). The results for the water contact angle in Figure 8c indicated a slight increase in the hydrophobicity of the silver-loaded composite films compared to the DAC@Ag0 film. Therefore, the loading of silver nanoparticles is beneficial in improving the resistance of the film to water [55].

In addition, the antimicrobial properties were tested by the disc diffusion method, and each sample was tested three times. After 24 h of incubation, the antimicrobial properties were determined by detecting the inhibition zones of the films against *E. coli* and *S. aureus*. Figure 8d and Figure 9 show the inhibition behavior of the film materials against *E. coli* and *S. aureus*, in which the DAC@Ag0 films had no inhibition zones and the silver-loaded composite films showed excellent inhibition against both representative bacteria. This was the result of the destruction of the cell membranes of the bacteria by AgNPs in the silver-loaded films, causing the inactivity and death of bacteria. In addition, the width of the inhibition circle enlarged with the increase in nanosilver content, indicating that the antibacterial properties gradually improved with the increase in nanosilver content. In Figure 8d, the antibacterial activity of these films was stronger against *S. aureus* compared to *E. coli*, which was attributed to the difference in the cell wall structure of bacteria that made it easier for AgNPs to enter Gram-positive bacteria than Gram-negative bacteria [42].

## 3. Experimental Section

### 3.1. Materials and Chemicals

Microcrystalline cellulose (MCC) was purchased from Sinopharm Chemical Reagent Co., Ltd. (Shanghai, China), sodium periodate (NaIO_4_, purity ≥ 99%) was purchased from Aladdin Biochemical Science and Technology Co., Ltd. (Shanghai, China), and ethylene glycol (purity ≥ 98%) was purchased from Shanghai Runjie Chemical Reagent Co. Silver nitrate (AgNO_3_, purity ≥ 99%) and aqueous ammonia (purity ≥ 25%) were provided by Guangzhou Chemical Reagent Factory (Guangzhou, China), and all chemicals were used as is without further purification. The water used in the experiments was deionized.

### 3.2. Periodate Oxidation and Hydrothermal Dissolution of Cellulose

Twelve grams of MCC and 500 mL of deionized water were added to a 1000 mL blue-capped bottle to form a suspension. Then, 19.0125 g of sodium periodate (molar ratio of NaIO_4_ to dehydrated glucose units AGU of 1.2) was weighed and dissolved in 242 mL of deionized water. Finally, the sodium periodate solution was added to the above suspension, and the bottle was completely wrapped with aluminum foil to prevent exposure of sodium periodate to light. The bottle was placed in an oil bath at 48 °C and stirred at 800 rpm. After 19 h, an excess of ethylene glycol was added to terminate the reaction. Then, the suspensions were left to stratify, and the supernatant was poured off. The lower solid was washed and dispersed in deionized water to obtain suspensions with a concentration of 5 wt%. The suspensions were then heated for 45–60 min with magnetic stirring (1000 rpm) at 100 °C. After the solid was dissolved, the hot solution was cooled and centrifuged to remove a small amount of insoluble material to obtain dialdehyde cellulose aqueous solution (DAC).

### 3.3. Preparation of AgNPs

First, 50 g of 0.5 wt% aqueous silver nitrate solution was prepared, and then roughly 0.75 g of 25 wt% ammonia was added to the prepared 50 g of aqueous silver nitrate solution to obtain silver ammonia solution ([Ag(NH_3_)_2_]OH). Next, a portion of the aqueous DAC solution was diluted to 0.1 wt% and then mixed with the silver ammonia solution and reacted under heating and stirring in an oil bath. The reaction originated from the classical Tollens reaction. After the reaction, the mixture was dialyzed off the unreacted DAC solution with a dialysis bag (cut-off molecular weight, 3500) to obtain pure silver nanoparticle colloid. A portion of the silver nanoparticle colloid was lyophilized at −40 °C for subsequent assay characterization. During the reaction process, the optimal conditions for the preparation of nanosilver were explored by controlling different mass ratios of DAC to silver ammonia solution, reaction temperature, heating time and different silver ion contents.

### 3.4. In Situ Synthesis of DAC@Ag Composite Films

Silver ammonia solutions of 0, 200 μL, 400 μL, 600 μL, 800 μL, and 1000 μL were added to 10 mL of 5 wt% DAC aqueous solution, respectively, then stirred magnetically at 50 °C for 60 min. The obtained solutions were each transferred to a Petri dish (diameter of 9 cm), air-dried naturally, and the obtained films were named DAC@Ag0, DAC@Ag1, DAC@Ag2, DAC@Ag3, DAC@Ag4, and DAC@Ag5, respectively.

### 3.5. Analytical Characterization

#### 3.5.1. Determination of DAC Oxidation

The degree of oxidation of DAC was calculated indirectly by the absorbance of sodium periodate consumption [28]. Briefly, after 19 h of oxidation (before the addition of ethylene glycol), an appropriate amount of liquid was removed from the mixture and diluted 40-fold, and then its absorbance at 290 nm was measured by UV spectrophotometer. The degree of oxidation (DO) was calculated by the following equation [56]:DO(%) = B/A × 100(1)
where A = moles of periodate consumed per dehydrated glucose unit (162) (1.00), and B = moles of periodate actually consumed per dehydrated glucose unit (162).

#### 3.5.2. Analytical Characterization of Nanosilver and Cellulose

The nanosilver was spectroscopically scanned with a UV spectrophotometer (UV-2600, Shimadzu, China, scanning range 800–300 nm, interval 0.5 nm, speed fast). The average size of nanosilver was examined by a Malvern laser particle sizer (3000HSA, Malvern, UK). The FTIR spectra of MCC and lyophilized DAC were obtained by the potassium bromide press method on a Fourier infrared spectrometer (FT-IR 4700, JASCO, Japan) with a scan range of 4000–400 cm^−1^ and a resolution of 2 cm^−1^. The nanosilver colloids were dropped on ultrathin carbon film copper grids, and the morphological characteristics of the nanosilver were observed by transmission electron microscopy (TEMJEM-1400 Plus, Japan) after it was air-dried. The X-ray photoelectron spectra (XPS) of the samples were recorded by an AXIS Ultra DLD spectrometer (Kratos, UK) with Mg Kα radiation (hγ = 1253.6 eV) as the excitation source in steps of 0.1 eV. The crystal structure of the samples was studied by a D8 advanced X-ray diffractometer (Bruker, Germany), where the Cu Kα filtered radiation (λ = 0.15418 nm) was generated at 40 kV and 40 mA current density. The samples were scanned at a scan rate of 2 °/s in steps of 0.0195° over a range of (2θ) from 5 to 90°. The data were then processed using Jade 6.5 software to obtain an approximate crystallinity of AgNPs, and the crystallinity of cellulose was calculated using the following equation [57]:CI (%) = [(I_200_ − I_am_)/I_200_] × 100(2)
where I_200_ is the maximum intensity corresponding to the main peak in plane (200) at 2θ = 22.6°, and I_am_ is the diffraction intensity in the amorphous region at 2θ = 18.5°. The average particle size of nanosilver (D) was calculated using the Debye–Scherrer equation [46]:D = Kλ/(βcosθ)(3)
where λ is the wavelength of X-ray diffraction (Cu Kα = 0.15418 nm), K is a constant, taken here as 0.89, β is the half-peak width, and θ is the diffraction angle.

#### 3.5.3. Analytical Characterization of the Composite Films

The surface and cross-sectional morphology of the composite films were observed with a field emission scanning electron microscope (SU5000, HITACHI, Japan) at an accelerating voltage of 3 kV, and were analyzed by energy dispersive X-ray spectroscopy (EDS) for elemental mapping (the cross-sectional samples were obtained by liquid nitrogen embrittlement). The infrared spectra of the composite films were obtained with a Fourier infrared spectrometer (FT-IR 4700, JASCO, Japan), setting a scan range of 4000–400 cm^−1^ and a resolution of 2 cm^−1^. To characterize the transmittance of the composite films, a UV spectrophotometer (UV-2600, Shimadzu, China) was used to determine the UV transmittance of the samples, setting the scan range from 800–300 nm with an interval of 0.5 nm and a fast speed.

The mechanical properties of the films were obtained by a universal testing machine (INSTRON 5565, USA) at a constant displacement rate of 5 mm/min at room temperature. The water vapor transmission rate (WVR) and oxygen transmission rate (OTR) of the films were tested by a water vapor transmission rate tester (W413 2.0, China) and an oxygen transmission rate tester (Y310 2.0, China), respectively, at a temperature of 30 °C and relative humidity of 75%. The water contact angle was measured and calculated on a contact angle analyzer (ZJ-7000, China): a drop of water (2 μL) was placed on the film surface using an automatic piston syringe and subsequently photographed to capture its side-view image, then the contact angle was calculated. The water stability of the films was measured by immersing the film materials in water for 1 h to measure the water absorption rate and recording the weight change of the sample before and after water immersion [58]. Water absorption rate = (W_2_ − W_1_)/W_1_, where W_1_ is the film mass before water immersion, and W_2_ is the film mass after water immersion. Each sample was tested three times and averaged.

In addition, the antibacterial performance of DAC-AgNP composite films was evaluated by disc diffusion method using *E. coli* and *S. aureus* as representative Gram-negative and Gram-positive bacteria, respectively. The strain and 0.9 wt% physiological saline were used to prepare an active bacterial suspension with a concentration of approximately 5 × 10^5^ cfu/mL. Subsequently, the film samples were sterilized with UV light for 30 min (samples were cut to 10 mm in diameter). About 10 mL of sterilized nutrient agar medium was poured into sterile Petri dishes. The medium was solidified and inoculated with 50 μL of bacterial suspension, then the samples were affixed to the medium and incubated at 37 °C for 24 h. After incubation, the Petri dishes were removed to measure the inhibition circle. The assay was repeated three times for all samples.

## 4. Conclusions

To sum up, in this study, silver nanoparticles were successfully prepared in a homogeneous aqueous solution of dialdehyde cellulose without adding other reducing and stabilizing agents. The prepared silver nanoparticles were mainly spherical in shape with an average size of about 25 nm under optimal conditions. Moreover, in situ-synthesized DAC@Ag composite films were also prepared. The obtained films were smooth and dense. The AgNPs were uniformly distributed on the surface and the inside of the films. Upon the increase in AgNP content, the tensile strength of the composite films rose to 94.6 MPa and then gradually decreased. The obtained composite films not only had good transparency but also had good UV-blocking properties. In addition, the composite films also had excellent oxygen and water vapor barrier, water stability, and outstanding antibacterial ability against *E. coli* as well as *S. aureus*. Therefore, this study not only developed a green and simple method for the preparation of metal nanoparticles but also provided a novel route for the in situ synthesis of antibacterial composite films. The synthesized nanosilver can be isolated for use in other fields, and the synthesized silver-loaded composite films can be further used in antimicrobial packaging materials.

## Figures and Tables

**Figure 1 molecules-28-02956-f001:**
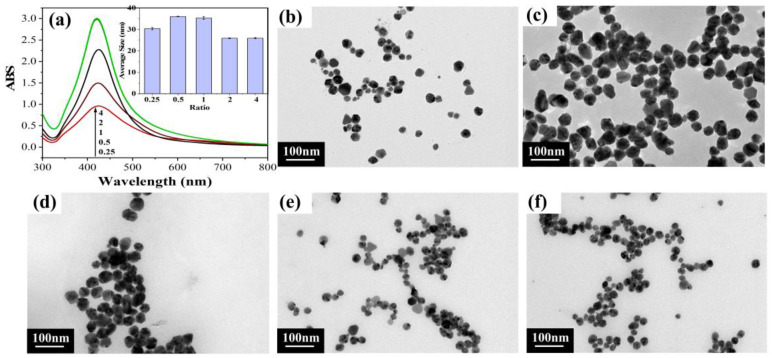
Results for different DAC to silver ammonia solution mass ratios: (**a**) UV-vis spectra and (inset) average size histogram; (**b**–**f**) TEM images of DAC to silver ammonia solution mass ratios of 0.25–4, respectively.

**Figure 2 molecules-28-02956-f002:**
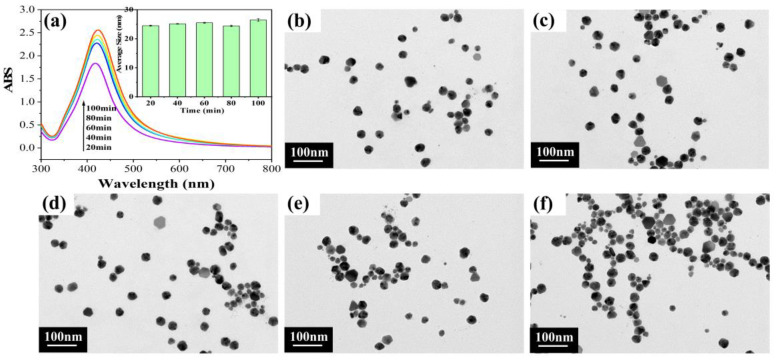
Results for different reaction times: (**a**) UV-vis spectra and (inset) average size histogram; (**b**–**f**) TEM images for reaction times of 20–100 min, respectively.

**Figure 3 molecules-28-02956-f003:**
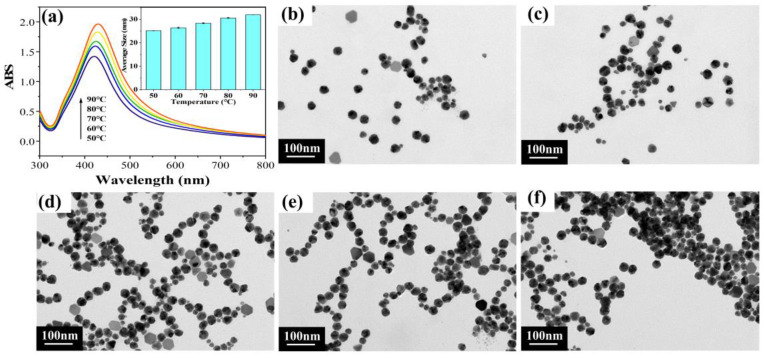
Results for different reaction temperatures: (**a**) UV-vis spectra and (inset) average size histogram; (**b**–**f**) TEM images for reaction temperatures of 50–90 °C, respectively.

**Figure 4 molecules-28-02956-f004:**
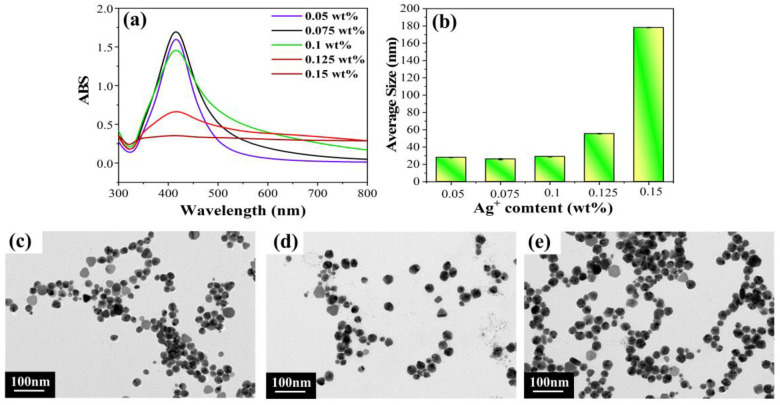
Results for different Ag^+^ contents: (**a**) UV-vis spectra; (**b**) average size histogram; (**c**–**e**) TEM images for Ag^+^ contents of 0.05 wt%–0.1 wt%, respectively.

**Figure 5 molecules-28-02956-f005:**
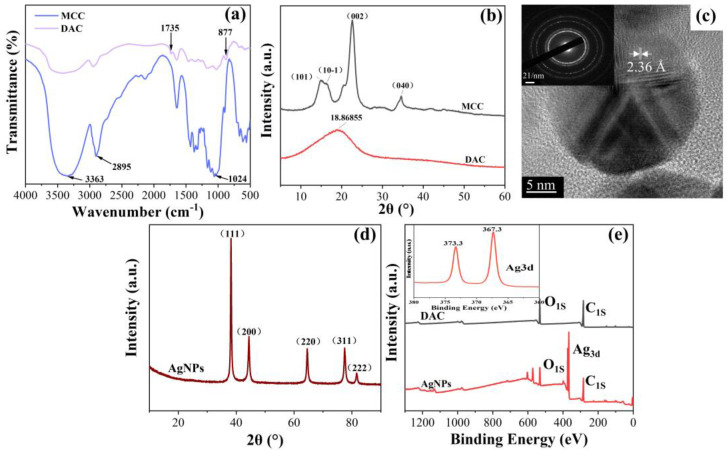
FT-IR spectra (**a**) and XRD spectra (**b**) of MCC and DAC, (**c**) in high resolution, (**c**, inset) selected area electron diffraction of AgNPs, (**d**) XRD spectra of AgNPs, (**e**) XPS spectra of DAC and AgNPs.

**Figure 6 molecules-28-02956-f006:**
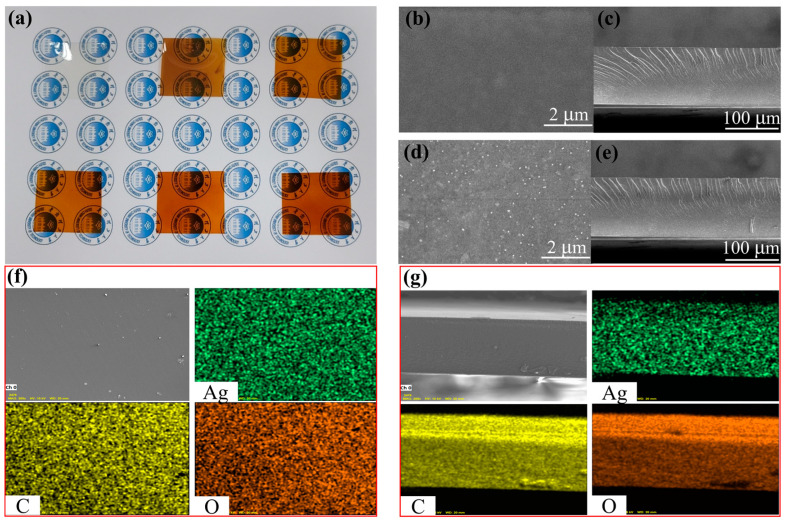
Optical photograph of DAC@Ag0-DAC@Ag5 films (**a**), SEM images of the plane and cross section of DAC@Ag0 film (**b**,**c**) and DAC@Ag1 film (**d**,**e**), EDS element mapping of composite film planes (**f**) and cross sections (**g**).

**Figure 7 molecules-28-02956-f007:**
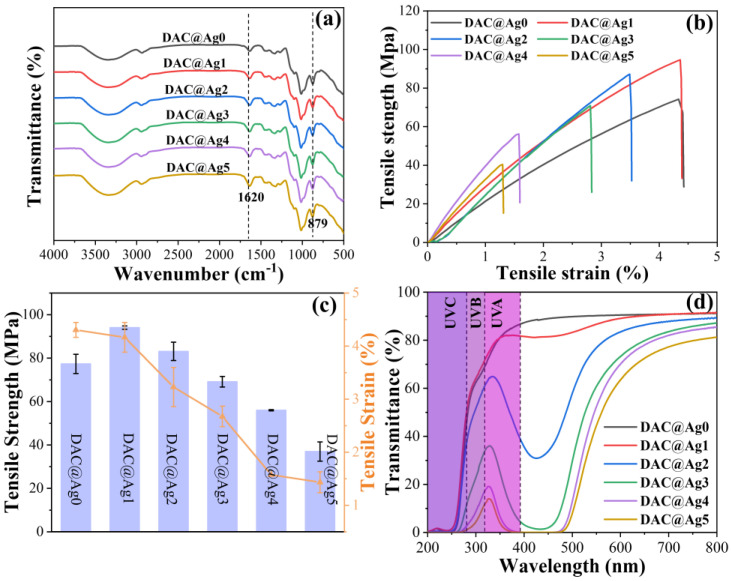
FT-IR spectra (**a**), mechanical properties (**b**,**c**) and UV transmission spectra (**d**) of DAC@Ag0-DAC@Ag5 films.

**Figure 8 molecules-28-02956-f008:**
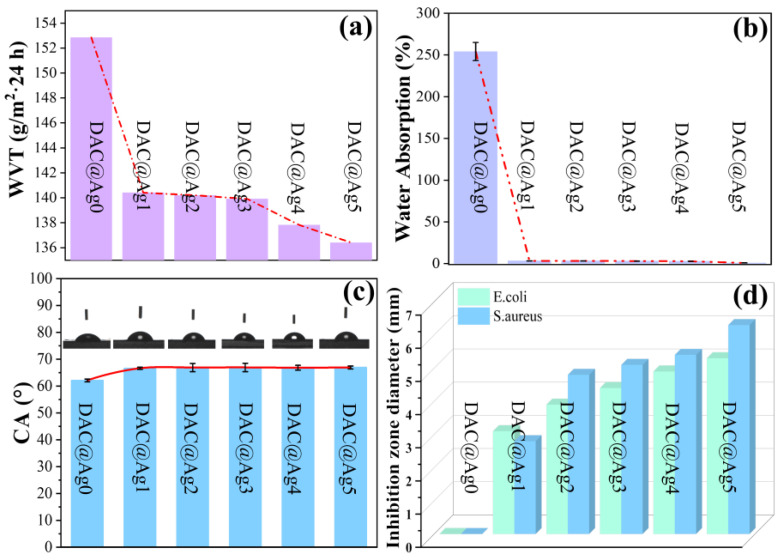
Water vapor transmission rate (WRT) (**a**), water absorption (**b**), water contact angle (**c**), and bacterial inhibition zone (**d**) of DAC@Ag0-DAC@Ag5 films.

**Figure 9 molecules-28-02956-f009:**
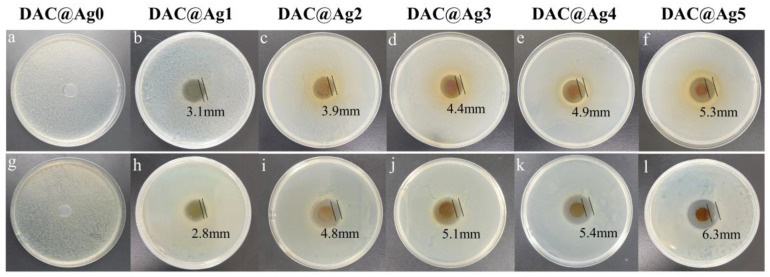
Antibacterial test of DAC@Ag0-DAC@Ag5 films. Inhibition images of the DAC@Ag0-DAC@Ag5 films against *E. coli* (**a**–**f**) and *S. aureus* (**g**–**l**).

**Table 1 molecules-28-02956-t001:** Tensile testing, antimicrobial properties, barrier and water stability results for DAC@Ag films.

Sample	Tensile Testing			Aw (°)	Inhibition Zone (mm)		WVT (g/m^2^·24 h)	ORT (cm^3^/m^2^·24 h·0.1 MPa)	Water Absorption (%)
	E (GPa)	σ (MPa)	ε (%)		*E. coli*	*S. aureus*			
DAC@Ag0	2.26	77.38	4.31	62.12	0	0	152.84	<0.02	254.12
DAC@Ag1	2.99	94.07	4.17	66.63	3.1	2.8	140.41	<0.02	3.31
DAC@Ag2	3.16	83.13	3.23	66.90	3.9	4.8	140.20	<0.02	3.24
DAC@Ag3	3.08	69.15	2.67	66.93	4.4	5.1	139.91	<0.02	3.05
DAC@Ag4	4.59	56.01	1.57	66.85	4.9	5.4	137.82	<0.02	2.83
DAC@Ag5	3.88	36.99	1.43	66.94	5.3	6.3	136.41	<0.02	0.81

## Data Availability

The data presented in this study are available on request from the corresponding author.

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
