# Peer review of "Dialdehyde Cellulose Solution as Reducing Agent: Preparation of Uniform Silver Nanoparticles and In Situ Synthesis of Antibacterial Composite Films with High Barrier Properties"

_molecules, 2023, doi:10.3390/molecules28072956_

Round 1

Reviewer 1 Report

The manuscript has been reviewed. Nice work has been performed but needs substantial changes:

1. Some corrections are highlighted in the pdf.

2. Polishing in the language required.

3. Authors should revise the conclusion including the analytical results.

4. If possible at some level, perform MIC and add.

5. Abstract needs improvement.

Reviewer 2 Report

This is a well-written manuscript, describing the synthesis of silver nanoparticles and a study of their antimicrobial properties.
Overall, the setting is highly interesting and the mild reduction method certainly gives new impulses to the field.

Overall, I red the manuscript with great pleasure and I reommend publication after some corrections:

Please in the whole manuscript re-check the English.

"with WVT and ORT of 136.41 g/m2·24h (30ºC,75%RH) and <0.02 cm3/m2·24h·0.1MPa"
--- between each number and the corresponding unit there must be a space.
The same applies to
"The silver ammonia solution of 0, 400uL, 800uL, 1200uL, 1600uL, and 2000uL was added to 10mL of 5wt% DAC aqueous solution, respectively, stirred magnetically at 50 °C for 20min, the obtained solution was transferred to a Petri dish (diameter of 9cm),"
and the rest of the manuscript

"but also have low cytotoxicity and no micro-bial resistance,12 which has expanded their applications in medicine13, antimicrobial pack-aging14, material coating 15, and water treatment16."
-- as far as I know, all references should be placed after the commas and the full stop, not before (compare also other sections of the manuscript ...).

"for synthesizing AgNPs with Excellent antibacterial property."
-- "excellent" with lower case letter e

"(AgNO3, purity ≥ 99%) and aqueous"
-- "3" in subscript

"Moreover, in-situ synthesized"
-- move the hyphen behind "situ"

In Figure 1(a) the writing is barely readable.
The same in Figure 2(a) and 3(a)

Figure 4(b)
-- here the X-axis has the unit "Ag+ content (Wt%)", but in the legend is describe as particle-size histogram (which would be in nm).
Correcct this

Reviewer 3 Report

The authors of the manuscript “Dialdehyde Cellulose Solution as Reducing agent: Preparation of Uniform Silver Nanoparticles and In-situ Synthesis of Anti-bacterial Composite Films with High Barrier” presented the physicochemical characterization of newly synthesized AgNPs, obtained after dialysis / lyophilization and using DAC as a reducing agent.

With all due respect, some aspects of the experimental protocol are quite unclear.

-          For instance, why the authors did not evaluate the antibacterial effects of sole AgNPs?

-          The characterization of Ag@DAC films confirmed the presence of metallic nanoparticles, but did the authors determine their compositional / structural / morphological features?

-          What are the differences between the lyophilized and air-dried Ag@DAC particles? Given the differences between the synthesis processes, are those characteristics the same as in the case of initial AgNPs?

-          Did the authors evaluate the stability of AgNPs or AgNPs-embedded films?

-          Also, did the author determine the release rate of the silver ions / nanoparticles from the composite films?

-          The Materials and Method section presents a better and complete description of the materials and systems used to obtain and characterizes the AgNPs and cellulose.

-          Both in the Abstract and at the end of the Introduction paragraph, it is specified that the simple Tollens reaction was used, but in the Materials and Methods section, nothing is specified about this process.

-          In Figure 2, what mass ratio of DAC to Ag+ was used?

-          I suggest introducing the Materials and Method section paraph with the Antibacterial Activity Study. What was the number of replicates for each antimicrobial assay?

-          How was the Cytotoxicity Study realized?

-          E. coli as well as Staphylococcus aureus - abbreviation at the first use in the text, and maintain the same abbreviation in the text. 

Round 2

Reviewer 3 Report

The authors of the manuscript entitled “Dialdehyde Cellulose Solution as Reducing agent: Preparation 2 of Uniform Silver Nanoparticles and In-situ Synthesis of Anti- 3 bacterial Composite Films with High Barrier” made modifications according to the suggestions and observations and provide also supplementary material and discussion along with discussions.

In this situation, the manuscript can be published in its current form.
